# INSTANCE CROSS ENTROPY FOR DEEP METRIC LEARNING

## ABSTRACT

Loss functions play a crucial role in deep metric learning thus a variety of them have been proposed. Some supervise the learning process by pairwise or triplet-wise similarity constraints while others take advantage of structured similarity information among multiple data points. In this work, we approach deep metric learning from a novel perspective. We propose instance cross entropy (ICE) which measures the difference between an estimated instance-level matching distribution and its ground-truth one. ICE has three main appealing properties. Firstly, similar to categorical cross entropy (CCE), ICE has clear probabilistic interpretation and exploits structured semantic similarity information for learning supervision. Secondly, ICE is scalable to infinite training data as it learns on mini-batches iteratively and is independent of the training set size. Thirdly, motivated by our relative weight analysis, seamless sample reweighting is incorporated. It rescales samples' gradients to control the differentiation degree over training examples instead of truncating them by sample mining. In addition to its simplicity and intuitiveness, extensive experiments on three real-world benchmarks demonstrate the superiority of ICE.

## 1 INTRODUCTION

Deep metric learning (DML) aims to learn a non-linear embedding function (a.k.a. distance metric) such that the semantic similarities over samples are well captured in the feature space (Tadmor et al., 2016; Sohn, 2016). Due to its fundamental function of learning discriminative representations, DML has diverse applications, such as image retrieval (Song et al., 2016), clustering (Song et al., 2017), verification (Schroff et al., 2015), few-shot learning (Vinyals et al., 2016) and zero-shot learning (Bucher et al., 2016).

A key to DML is to design an effective and efficient loss function for supervising the learning process, thus significant efforts have been made (Chopra et al., 2005; Schroff et al., 2015; Sohn, 2016; Song et al., 2016; 2017; Law et al., 2017; Wu et al., 2017). Some loss functions learn the embedding function from pairwise or triplet-wise relationship constraints (Chopra et al., 2005; Schroff et al., 2015; Tadmor et al., 2016). However, they are known to not only suffer from an increasing number of non-informative samples during training, but also incur considering only several instances per loss computation. Therefore, informative sample mining strategies are proposed (Schroff et al., 2015; Wu et al., 2017; Wang et al., 2019b). Recently, several methods consider semantic relations among multiple examples to exploit their similarity structure (Sohn, 2016; Song et al., 2016; 2017; Law et al., 2017). Consequently, these structured losses achieve better performance than pairwise and triple-wise approaches.

In this paper, we tackle the DML problem from a novel perspective. Specifically, we propose a novel loss function inspired by CCE. CCE is well-known in classification problems owing to the fact that it has an intuitive probabilistic interpretation and achieves great performance, e.g., ImageNet classification (Russakovsky et al., 2015). However, since CCE learns a decision function which predicts the class label of an input, it learns class-level centres for reference (Zhang et al., 2018; Wang et al., 2017a). Therefore, CCE is not scalable to infinite classes and cannot generalise well when it is directly applied to DML (Law et al., 2017).

With scalability and structured information in mind, we introduce instance cross entropy (ICE) for DML. It learns an embedding function by minimising the cross entropy between a predicted instance-level matching distribution and its corresponding ground-truth. In comparison with CCE,

given a query, CCE aims to maximise its *matching probability with the class-level context vector* (weight vector) of its ground-truth class, whereas ICE targets at maximising its *matching probability with it similar instances*. As ICE does not learn class-level context vectors, it is scalable to infinite training classes, which is an intrinsic demand of DML. Similar to (Sohn, 2016; Song et al., 2016; 2017; Law et al., 2017; Goldberger et al., 2005; Wu et al., 2018), ICE is a structured loss as it also considers all other instances in the mini-batch of a given query. We illustrate ICE with comparison to other structured losses in Figure 1.

A common challenge of instance-based losses is that many training examples become trivial as model improves. Therefore, we integrate seamless sample reweighting into ICE, which functions similarly with various sample mining schemes (Sohn, 2016; Schroff et al., 2015; Shi et al., 2016; Yuan et al., 2017; Wu et al., 2017). Existing mining methods require either separate time-consuming process, e.g., class mining (Sohn, 2016), or distance thresholds for data pruning (Schroff et al., 2015; Shi et al., 2016; Yuan et al., 2017; Wu et al., 2017). Instead, our reweighting scheme works without explicit data truncation and mining. It is motivated by the relative weight analysis between two examples. The current common practice of DML is to learn an angular embedding space by projecting all features to a unit hypersphere surface (Song et al., 2017; Law et al., 2017; Movshovitz-Attias et al., 2017). We identify the challenge that without sample mining, informative training examples cannot be differentiated and emphasised properly because the relative weight between two samples is strictly bounded. We address it by sample reweighting, which rescales samples' gradient to control the differentiation degree among them.

Finally, for intraclass compactness and interclass separability, most methods (Schroff et al., 2015; Song et al., 2016; Tadmor et al., 2016; Wu et al., 2017) use distance thresholds to decrease intraclass variances and increase interclass distances. In contrast, we achieve the target from *a perspective of instance-level matching probability*. *Without any distance margin constraint*, ICE makes no assumptions about the boundaries between different classes. Therefore, ICE is easier to apply in applications where we have no prior knowledge about intraclass variances.

Our contributions are summarised: (1) We approach DML from a novel perspective by taking in the key idea of matching probability in CCE. We introduce ICE, which is scalable to an infinite number of training classes and exploits structured information for learning supervision. (2) A seamless sample reweighting scheme is derived for ICE to address the challenge of learning an embedding subspace by projecting all features to a unit hypersphere surface. (3) We show the superiority of ICE by comparing with state-of-the-art methods on three real-world datasets.

## 2 RELATED WORK

### 2.1 STRUCTURED LOSSES BY QUERY VERSUS CLASS CENTRES

**Heated-up**, **NormFace**, **TADAM**, **DRPR**, **Prototypical Networks**, **Proxy-NCA**. These methods calculate the similarities between a query and class centres (a.k.a. proxies or prototypes) instead of other instances (Zhang et al., 2018; Wang et al., 2017a; Oreshkin et al., 2018; Law et al., 2019; Snell et al., 2017; Movshovitz-Attias et al., 2017). In Heated-up and NormFace, the class centres are learned parameters of a fully connected layer, which is similar to Center Loss (Wen et al., 2016). While in TADAM, DRPR, and Prototypical Networks, a class centre is the mean over all embeddings of a class. By comparing a sample with other examples other than class centres, more informative instances can contribute more in ICE.

### 2.2 STRUCTURED LOSSES BY QUERY VERSUS INSTANCES

**NCA** (Goldberger et al., 2005), **S-NCA** (Wu et al., 2018). NCA learns similarity relationships between instances. Since original NCA learns the whole training data and its time complexity is quadratically proportional to the scale of training data, S-NCA is proposed recently with linear time complexity with respect to the training data size. Instead, ICE is scalable to infinite training data by iterative learning on randomly sampled small-scale instances matching tasks. S-NCA and NCA share the same learning objective. However, they treat the event of all similar instance being correctly recognised *as a whole* by a sum accumulator. Instead, we maximise the probability of every similar sample being correctly identified *individually*. Therefore, ICE's optimisation task is harder, leading to better generalisation.

**N-pair-mc** (Sohn, 2016). The aim of *N*-pair-mc is to identify one positive example from $N-1$ negative examples of $N-1$ classes (one negative example per class). In other words, only one

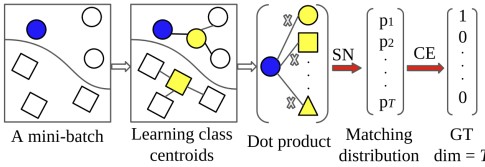

(a) A query versus **learned parametric class centroids**. All $T$ classes in the training set are considered. Prior work: CCE, Heated-up (Zhang et al., 2018), NormFace (Wang et al., 2017a).

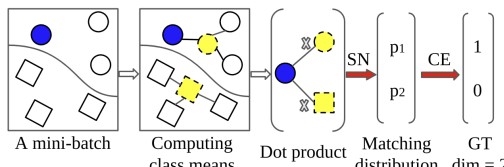

(b) A query versus **non-parametric class means**. Only classes in the mini-batch are considered. Representative work: TADAM (Oreshkin et al., 2018), DRPR (Law et al., 2019), Prototypical Networks (Snell et al., 2017).

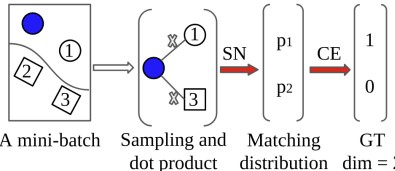

(c) *N*-pair-mc (Sohn, 2016): A query versus **one instance per class**. A mini-batch has to be 2 examples per class rigidly. Only one instance per negative class is randomly sampled out of 2.

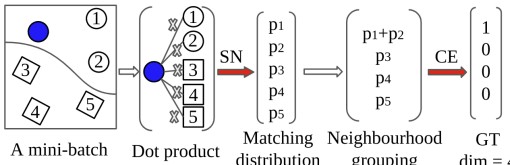

(d) NCA (Goldberger et al., 2005) and S-NCA (Wu et al., 2018): A query versus **the rest instances**.

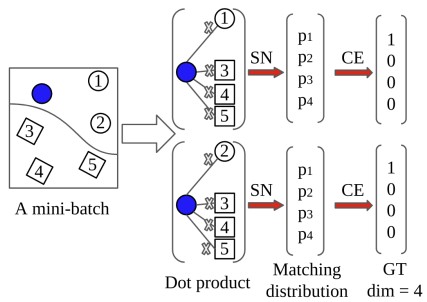

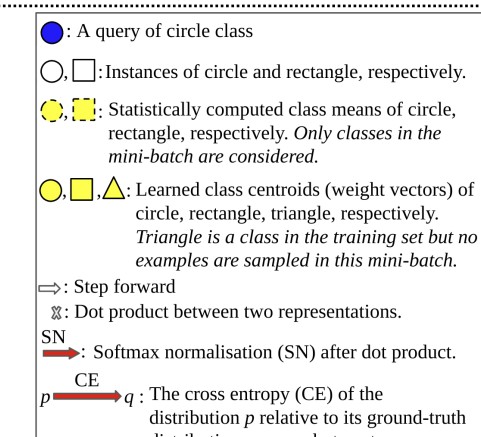

(e) Our ICE: A query versus **one positive and all negatives per distribution**. A query's number of matching distributions is defined by the number of its positive examples.

Figure 1: Our ICE and related losses. The first row shows prior work of a query versus class centres/means while the second row displays the work of a query versus instances. Note that the cross entropy computation and interpretation are different in different losses. For a mini-batch, we show two classes, i.e., circle and rectangle, with 3 examples per class except *N*-pair-mc which requires 2 samples per class. The icons are at the right bottom. GT means ground-truth matching distribution. When illustrating the losses of a query versus instances in (c), (d) and (e), we index those instances with numbers for clarity, except the query.

positive and one negative instance per class are considered per loss computation by simulating CCE exactly. Instead, ICE exploits all negative examples to benefit from richer information. When constructing mini-batches, *N*-pair-mc requires expensive offline class mining and samples 2 images per class. According to (Sohn, 2016) *N*-pair-mc is superior to NCA.

**Hyperbolic** (Nickel & Kiela, 2018). It aims to preserve the similarity structures among instances as well. However, it learns a hyperbolic embedding space where the distance depends only on norm of embeddings. Instead, we learn an angular space where the similarity depends only on the angle between embeddings. Besides, Hyperbolic requires a separate sampling of semantic subtrees when the dataset is large.

### 2.3 SAMPLE MINING AND WEIGHTING

Mining informative examples or emphasising on them are popular strategies in DML: 1) Mining non-trivial samples during training is crucial for faster convergence and better performance.

Therefore, sample mining is widely studied in the literature. In pairwise or triplet-wise approaches (Schroff et al., 2015; Wu et al., 2017; Huang et al., 2016; Yuan et al., 2017), data pairs with higher losses are emphasized during gradient backpropagation. As for structured losses, Lifted Struct (Song et al., 2016) also focuses on harder examples. Furthermore, (Sohn, 2016) and (Suh et al., 2019) propose to mine hard negative classes to construct informative input mini-batches. Proxy-NCA (Movshovitz-Attias et al., 2017) addresses the sampling problem by learning class proxies. 2) Assigning higher weights to informative examples is another effective scheme (Wang et al., 2019a;c). Beyond, there are some other novel perspectives to address sample mining or weighting, e.g., hardness-aware examples generation (Zheng et al., 2019) and divide-and-conquer of the embedding space (Sanakoyeu et al., 2019).

Our proposed ICE has a similarity scaling factor which helps to emphasise more on informative examples. Moreover, as described in (Schroff et al., 2015), very hard negative pairs are likely to be outliers and it is safer to mine semi-hard ones. In ICE, the similarity scaling factor is flexible in that it controls the emphasis degree on harder samples. Therefore, a proper similarity scaling factor can help mine informative examples and alleviate the disturbance of outliers simultaneously. *What makes ours different is that we do not heuristically design the mining or weighting scheme*. Instead, it is built-in and we simply scale it as demonstrated in Section 3.4.

## 2.4 DISCUSSION

We remark that Prototypical Networks, Matching Networks (Vinyals et al., 2016) and NCA are also scalable and do not require distance thresholds. Therefore, they are illustrated and differentiated in Figure 1. Matching Networks are designed specifically for one-shot learning. Similarly, (Triantafillou et al., 2017) design mAP-SSVM and mAP-DLM for few-shot learning, which directly optimises the retrieval performance mAP when multiple positives exist. FastAP (Cakir et al., 2019) is similar to (Triantafillou et al., 2017) and optimises the ranked-based average precision. Instead, ICE processes one positive at a time. Beyond, the setting of few-shot learning is different from deep metric learning: Each mini-batch is a complete subtask and contains a support set as training data and a query set as validation data in few-shot learning. Few-shot learning applies episodic training in practice.

Remarkably, TADAM formulates instances versus class centres and also has a metric scaling parameter for adjusting the impact of different class centres. Contrastively, ICE adjusts the influence of other instances. Furthermore, ours is not exactly distance metric scaling since we simply apply naive cosine similarity as the distance metric at the testing stage. That is why we interpret it as a weighting scheme during training.

## 3 INSTANCE CROSS ENTROPY

**Notation**. $\mathbf{X} = \{(\mathbf{x}_i, y_i)\}_{i=1}^N = \{\{\mathbf{x}_i^c\}_{i=1}^{N_c}\}_{c=1}^C$ is an input mini-batch, where $\mathbf{x}_i \in \mathbb{R}^{h \times w \times 3}$ and $y_i \in \{1, ..., C\}$ represent $i$-th image and the corresponding label, respectively; $\{\mathbf{x}_i^c\}_{i=1}^{N_c}$ is a set of $N_c$ images belonging to $c$-th class, $\forall c, N_c \geq 2$. The number of classes $C$ is generally much smaller than the total number of classes $T$ in the training set ($C \ll T$). Note that $T$ is allowed to be extremely large in DML. Given a sufficient number of different mini-batches, our goal is to learn an embedding function $f$ that captures the semantic similarities among samples in the feature space. We represent deep embeddings of X as $\{\{\mathbf{f}_i^c = f(\mathbf{x}_i^c)\}_{i=1}^{N_c}\}_{c=1}^C$. Given a query, 'positives' and 'negatives' refer to samples of the same class and different classes, respectively.

### 3.1 REVISITING CATEGORICAL CROSS ENTROPY

CCE is widely used in a variety of tasks, especially classification problems. As demonstrated in (Liu et al., 2016), a deep classifier consists of two joint components: *deep feature learning* and *linear classifier learning*. The feature learning module is a transformation (i.e., embedding function $f$) composed of convolutional and non-linear activation layers. The classifier learning module has one neural layer, which learns $T$ class-level context vectors such that any image has the highest compatibility (logit) with its ground-truth class context vector:

$$p(\mathbf{w}_{y_i}|\mathbf{x}_i) = \frac{\exp(\mathbf{f}_i^\top \mathbf{w}_{y_i})}{\sum_{k=1}^T \exp(\mathbf{f}_i^\top \mathbf{w}_k)} \quad \text{and} \quad L_{\text{CCE}}(\mathbf{X}; f, \mathbf{W}) = -\sum_{i=1}^N \log p(\mathbf{w}_{y_i}|\mathbf{x}_i), \quad (1)$$

where $\mathbf{f}_i = f(\mathbf{x}_i) \in \mathbb{R}^d$ is a $d$-dimensional vector, $p(\mathbf{w}_{y_i}|\mathbf{x}_i)$ is the probability (softmax normalised logit) of $\mathbf{x}_i$ matching $\mathbf{w}_{y_i}$, $\mathbf{W} = \{\mathbf{w}_k \in \mathbb{R}^d\}_{k=1}^T$ is the learned parameters of the classifier. During training, the goal is to maximise the joint probability of all instances being correctly classified. The identical form is to minimise the negative log-likelihood, i.e., $L_{\text{CCE}}(\mathbf{X}; f, \mathbf{W})$. Therefore, the learning objective of CCE is:

$$\arg\max_{f,\mathbf{W}} \prod_{i=1}^N p(\mathbf{w}_{y_i}|\mathbf{x}_i) = \arg\min_{f,\mathbf{W}} L_{\text{CCE}}(\mathbf{X}; f, \mathbf{W}). \tag{2}$$

## 3.2 INSTANCE CROSS ENTROPY

In contrast to CCE, ICE is a loss for measuring instance matching quality (lower ICE means higher quality) and does not need class-level context vectors. We remark that an anchor may have multiple positives, which are isolated in separate matching distributions. There is a matching distribution for every anchor-positive pair versus their negatives as displayed in Figure 1e.

Let $\mathbf{f}_a^c$ be a random query, we compute its similarities with the remaining points using dot product. We define the probability of the given anchor $\mathbf{x}_a^c$ matching one of its positives $\mathbf{x}_i^c (i \neq a)$ as follows:

$$p(\mathbf{x}_i^c|\mathbf{x}_a^c) = \frac{\exp(\mathbf{f}_a^{c\top}\mathbf{f}_i^c)}{\exp(\mathbf{f}_a^{c\top}\mathbf{f}_i^c) + \sum_{o \neq c}\sum_j \exp(\mathbf{f}_a^{c\top}\mathbf{f}_j^o)}, \tag{3}$$

where $\mathbf{f}_a^{c\top}\mathbf{f}_i^c$ is the similarity between $\mathbf{x}_a^c$ and $\mathbf{x}_i^c$ in the embedding space, $\sum_{o \neq c}\sum_j \exp(\mathbf{f}_a^{c\top}\mathbf{f}_j^o)$ is the sum of similarities between $\mathbf{x}_a^c$ and its all negatives. Similarly, when the positive is $\mathbf{x}_i^c$, the probability of one negative point $\mathbf{x}_j^o (o \neq c)$ matching the anchor is:

$$p(\mathbf{x}_j^o|\mathbf{x}_a^c, \mathbf{x}_i^c) = \frac{\exp(\mathbf{f}_a^{c\top}\mathbf{f}_j^o)}{\exp(\mathbf{f}_a^{c\top}\mathbf{f}_i^c) + \sum_{o \neq c}\sum_j \exp(\mathbf{f}_a^{c\top}\mathbf{f}_j^o)}. \tag{4}$$

We remark: (1) Dot product measures the similarity between two vectors; (2) Eq. (3) represents the probability of a query matching a positive while Eq. (1) is the probability of a query matching its ground-truth class. To maximise $p(\mathbf{x}_i^c|\mathbf{x}_a^c)$ and minimise $p(\mathbf{x}_j^o|\mathbf{x}_a^c, \mathbf{x}_i^c)$ simultaneously, we minimise the Kullback-Leibler divergence (Kullback & Leibler, 1951) between the predicted and ground-truth distributions, which is equivalent to minimise their cross entropy. Since the ground-truth distribution is one-hot encoded, the cross-entropy is $-\log p(\mathbf{x}_i^c|\mathbf{x}_a^c)$.

To be more general, for the given anchor $\mathbf{x}_a^c$, there may exist multiple matching points when $N_c > 2$, i.e., $|\{\mathbf{x}_i^c\}_{i \neq a}| = N_c - 1 > 1$. In this case, we predict one matching distribution per positive point. Our goal is to maximise the joint probability of all positive instances being correctly identified, i.e., $p_{\mathbf{x}_a^c} = \prod_{i \neq a} p(\mathbf{x}_i^c|\mathbf{x}_a^c)$. A case of two positives matching a given query is described in Figure 1e.

In terms of mini-batch, each image in $\mathbf{X}$ serves as the anchor iteratively and we aim to maximise the joint probability of all queries $\{\{p_{\mathbf{x}_a^c}\}_{a=1}^{N_c}\}_{c=1}^C$. Equivalently, we can achieve this by minimising the sum of all negative log-likelihoods. Therefore, our proposed ICE on $\mathbf{X}$ is as follows:

$$L_{\text{ICE}}(\mathbf{X}; f) = -\sum_{c=1}^C \sum_{a=1}^{N_c} \log p_{\mathbf{x}_a^c} = -\sum_{c=1}^C \sum_{a=1}^{N_c} \sum_{i \neq a} \log p(\mathbf{x}_i^c|\mathbf{x}_a^c). \tag{5}$$

## 3.3 REGULARISATION BY $L_2$ FEATURE NORMALISATION

Following the common practice in existing DML methods, we apply $L_2$-normalisation to feature embeddings before the inner product. Therefore, the inner product denotes the cosine similarity.

The similarity between two feature vectors is determined by their norms and the angle between them. Without $L_2$ normalisation, the feature norm can be very large, making the model training unstable and difficult. With $L_2$ normalisation, *all features are projected to a unit hypersphere surface*. Consequently, the semantic similarity score is merely determined by the direction of learned representations. Therefore, $L_2$ normalisation can be regarded as a regulariser during training[1]. Note that the principle is quite different from recent hyperspherical learning methods (Liu et al., 2017a; Wang et al., 2018b;a; Liu et al., 2017b; 2018b;a). They enforce the learned *weight parameters* to a unit hypersphere surface and diversify their angles. In contrast, feature normalisation is *output*

---

[1]The training without $L_2$ feature normalisation leads to the norm of features becoming very large easily and the dot product becoming INF.

*regularisation* and invariant to the parametrisation of the underlying neural network (Pereyra et al., 2017). In summary, our learning objective is:

$$\arg\max_f \prod_{c=1}^{C} \prod_{a=1}^{N_c} p_{\mathbf{x}_a^c} = \arg\min_f L_{\text{ICE}}(\mathbf{X}; f) \quad s.t. \quad \forall a, c, ||\mathbf{f}_a^c||_2 = 1. \tag{6}$$

The feature $L_2$-normalisation layer is implemented according to Wang et al. (2017a). It is a differentiable layer and can be easily inserted at the output of a neural net.

### 3.4 SAMPLE REWEIGHTING OF ICE

**Intrinsic sample weighting.** We find that ICE emphasises more on harder samples from the perspective of gradient magnitude. We demonstrate this by deriving the partial derivatives of $L_{\text{ICE}}(\mathbf{X}; f)$ with respect to positive and negative examples.

Given the query $\mathbf{x}_a^c$, the partial derivative of its any positive instance is derived by the chain rule:

$$\frac{\partial L_{\text{ICE}}(\mathbf{X}; f)}{\partial \mathbf{f}_i^c} = -\frac{\mathbf{f}_a^c \cdot \sum_{o \neq c} \sum_j \exp(\mathbf{f}_a^{c\top} \mathbf{f}_j^o)}{\exp(\mathbf{f}_a^{c\top} \mathbf{f}_i^c) + \sum_{o \neq c} \sum_j \exp(\mathbf{f}_a^{c\top} \mathbf{f}_j^o)} \quad = -\mathbf{f}_a^c \cdot (1 - p(\mathbf{x}_i^c | \mathbf{x}_a^c)). \tag{7}$$

Since $||\mathbf{f}_a^c||_2 = 1$, $w_{(\mathbf{x}_i^c; \mathbf{x}_a^c)} = ||\frac{\partial L_{\text{ICE}}(\mathbf{X}; f)}{\partial \mathbf{f}_i^c}||_2 = (1 - p(\mathbf{x}_i^c | \mathbf{x}_a^c))$ can be viewed as the weight of $\mathbf{f}_i^c$ when the anchor is $\mathbf{x}_a^c$. Thus, *ICE focuses more on harder positive samples*, whose $p(\mathbf{x}_i^c | \mathbf{x}_a^c)$ is lower.

Similarly, the partial derivative of its any negative sample is:

$$\frac{\partial L_{\text{ICE}}(\mathbf{X}; f)}{\partial \mathbf{f}_j^o} = \sum_{i \neq a} \frac{\mathbf{f}_a^c \cdot \exp(\mathbf{f}_a^{c\top} \mathbf{f}_j^o)}{\exp(\mathbf{f}_a^{c\top} \mathbf{f}_i^c) + \sum_{o \neq c} \sum_j \exp(\mathbf{f}_a^{c\top} \mathbf{f}_j^o)} \quad = \mathbf{f}_a^c \cdot \sum_{i \neq a} p(\mathbf{x}_j^o | \mathbf{x}_a^c, \mathbf{x}_i^c), \tag{8}$$

where $p(\mathbf{x}_j^o | \mathbf{x}_a^c, \mathbf{x}_i^c)$ is the matching probability between $\mathbf{x}_j^o$ and $\mathbf{x}_a^c$ given that the ground-truth example is $\mathbf{x}_i^c$. The weight of $\mathbf{x}_j^o$ w.r.t. $\mathbf{x}_a^c$ is: $w_{(\mathbf{x}_j^o; \mathbf{x}_a^c)} = ||\frac{\partial L_{\text{ICE}}(\mathbf{X}; f)}{\partial \mathbf{f}_j^o}||_2 = \sum_{i \neq a} p(\mathbf{x}_j^o | \mathbf{x}_a^c, \mathbf{x}_i^c)$. Clearly, *the harder negative samples own higher matching probabilities and weights*.

**Relative weight analysis.** In general, the relative weight (Tabachnick et al., 2007) is more notable as the exact weight will be rescaled during training, e.g., linear post-processing by multiplying the learning rate. Therefore, we analyse the relative weight between two positive points of the same anchor ($i \neq k \neq a$):

$$\frac{w_{(\mathbf{x}_i^c; \mathbf{x}_a^c)}}{w_{(\mathbf{x}_k^c; \mathbf{x}_a^c)}} = \frac{1 - p(\mathbf{x}_i^c | \mathbf{x}_a^c)}{1 - p(\mathbf{x}_k^c | \mathbf{x}_a^c)} \quad = \frac{\exp(\mathbf{f}_a^{c\top} \mathbf{f}_k^c) + \sum_{o \neq c} \sum_j \exp(\mathbf{f}_a^{c\top} \mathbf{f}_j^o)}{\exp(\mathbf{f}_a^{c\top} \mathbf{f}_i^c) + \sum_{o \neq c} \sum_j \exp(\mathbf{f}_a^{c\top} \mathbf{f}_j^o)}. \tag{9}$$

Similarly, the relative weight between two negative points of the same anchor ($o \neq c, l \neq c$) is:

$$\frac{w_{(\mathbf{x}_j^o; \mathbf{x}_a^c)}}{w_{(\mathbf{x}_k^l; \mathbf{x}_a^c)}} = \frac{\sum_{i \neq a} p(\mathbf{x}_j^o | \mathbf{x}_a^c, \mathbf{x}_i^c)}{\sum_{i \neq a} p(\mathbf{x}_k^l | \mathbf{x}_a^c, \mathbf{x}_i^c)} \quad = \frac{\exp(\mathbf{f}_a^{c\top} \mathbf{f}_j^o)}{\exp(\mathbf{f}_a^{c\top} \mathbf{f}_k^l)}. \tag{10}$$

Note that the positive relative weight in Eq. (9) is *only decided* by $\mathbf{f}_a^{c\top} \mathbf{f}_i^c$ and $\mathbf{f}_a^{c\top} \mathbf{f}_k^c$ while the negative relative weight in Eq. (10) is *only determined* by $\mathbf{f}_a^{c\top} \mathbf{f}_j^o$ and $\mathbf{f}_a^{c\top} \mathbf{f}_k^l$. The relative weight is merely determined by the dot product, which is in the range of $[-1, 1]$ and strictly bounded.

**Non-linear scaling for controlling the relative weight.** Inspired by (Hinton et al., 2015), we introduce a scaling parameter to modify the absolute weight non-linearly:

$$\hat{w}_{(\mathbf{x}_i^c; \mathbf{x}_a^c)} = \frac{\sum_{o \neq c} \sum_j \exp(s \cdot \mathbf{f}_a^{c\top} \mathbf{f}_j^o)}{\exp(s \cdot \mathbf{f}_a^{c\top} \mathbf{f}_i^c) + \sum_{o \neq c} \sum_j \exp(s \cdot \mathbf{f}_a^{c\top} \mathbf{f}_j^o)} \quad = 1 - \hat{p}(\mathbf{x}_i^c | \mathbf{x}_a^c), \tag{11}$$

$$\hat{w}_{(\mathbf{x}_j^o; \mathbf{x}_a^c)} = \sum_{i \neq a} \frac{\exp(s \cdot \mathbf{f}_a^{c\top} \mathbf{f}_j^o)}{\exp(s \cdot \mathbf{f}_a^{c\top} \mathbf{f}_i^c) + \sum_{o \neq c} \sum_j \exp(s \cdot \mathbf{f}_a^{c\top} \mathbf{f}_j^o)} \quad = \sum_{i \neq a} \hat{p}(\mathbf{x}_j^o | \mathbf{x}_a^c, \mathbf{x}_i^c), \tag{12}$$

where $s \geq 1$ is the scaling parameter. In contrast to $p$ and $w$, $\hat{p}$ and $\hat{w}$ represent the rescaled matching probability and partial derivative weight, respectively. We remark that we scale the absolute weight

---

**Algorithm 1** Learn by minimising ICE stochastically

---

**Batch setting**: $C$ classes, $N_c$ images from $c$-th class, batch size $N = \sum_{c=1}^{C} N_c$.
**Hyper-setting**: The scaling parameter $s$ and the number of iterations $\tau$.
**Input**: Initialised embedding function $f$, iteration counter $iter = 0$.
**Output**: Updated $f$.
**for** $iter < \tau$ **do**
   $iter = iter + 1$.
   Sample one mini-batch randomly $\mathbf{X} = \{\{\mathbf{x}_i^c\}_{i=1}^{N_c}\}_{c=1}^{C}$.
   **Step 1**: Feedforward $\mathbf{X}$ into $f$ to obtain feature representations $\{\{\mathbf{f}_i^c\}_{i=1}^{N_c}\}_{c=1}^{C}$.
   **Step 2**: Compute the similarities between an anchor and the remaining instances. Every example serves as the anchor iteratively.
   **for** $\mathbf{f}_a^c \in \{\{\mathbf{f}_i^c\}_{i=1}^{N_c}\}_{c=1}^{C}$ **do**
     **for** $\mathbf{f}_i^c \in \{\mathbf{f}_i^c\}_{i \neq a}$ **do**
       Compute $p(\mathbf{x}_i^c | \mathbf{x}_a^c)$ using Eq. (3). // We do not need to compute Eq. (4).
     **end for**
   **end for**
   Compute $L_{\text{ICE}}(\mathbf{X}; f)$ using Eq. (5).
   **Step 3**: Gradient back-propagation to update the parameters of $f$ using Eq. (15).
**end for**

---

non-linearly, which is an indirect way of controlling the relative weight. We do not modify the relative weight directly and Eq. (9) and Eq. (10) are only for introducing our motivation.

Our objective is to maximise an anchor's matching probability with its any positive instance competing against its negative set. Therefore, we normalise the rescaled weights based on each anchor:

$$\bar{w}_{(\mathbf{x}_i^c; \mathbf{x}_a^c)} = \frac{1}{N} \cdot \frac{\hat{w}_{(\mathbf{x}_i^c; \mathbf{x}_a^c)}}{\sum_{i \neq a} \hat{w}_{(\mathbf{x}_i^c; \mathbf{x}_a^c)} + \sum_{o \neq c} \sum_j \hat{w}_{(\mathbf{x}_j^o; \mathbf{x}_a^c)}} = \frac{1}{2N} \cdot \frac{1 - \hat{p}(\mathbf{x}_i^c | \mathbf{x}_a^c)}{\sum_{i \neq a}(1 - \hat{p}(\mathbf{x}_i^c | \mathbf{x}_a^c))}, \quad (13)$$

$$\bar{w}_{(\mathbf{x}_j^o; \mathbf{x}_a^c)} = \frac{1}{N} \cdot \frac{\hat{w}_{(\mathbf{x}_j^o; \mathbf{x}_a^c)}}{\sum_{i \neq a} \hat{w}_{(\mathbf{x}_i^c; \mathbf{x}_a^c)} + \sum_{o \neq c} \sum_j \hat{w}_{(\mathbf{x}_j^o; \mathbf{x}_a^c)}} = \frac{1}{2N} \cdot \frac{\sum_{i \neq a} \hat{p}(\mathbf{x}_j^o | \mathbf{x}_a^c, \mathbf{x}_i^c)}{\sum_{i \neq a}(1 - \hat{p}(\mathbf{x}_i^c | \mathbf{x}_a^c))}. \quad (14)$$

Note that the denominators in Eq. (13) and (14) are the accumulated weights of positives and negatives w.r.t. $\mathbf{x}_a^c$, respectively. *Although there are much more negatives than positives, the negative set and positive set contribute equally as a whole, as indicated by* $1/2$. $N = \sum_{c=1}^{C} N_c$ is the total number of instances in $\mathbf{X}$. We select each instance as the anchor iteratively and treat all anchors equally, as indicated by $1/N$.

It is worth noting that during back-propagation, the magnitudes of partial derivatives in Eq. (7) and Eq. (8), i.e., $w_{(\mathbf{x}_i^c; \mathbf{x}_a^c)}$ and $w_{(\mathbf{x}_i^c; \mathbf{x}_a^c)}$, are replaced by $\bar{w}_{(\mathbf{x}_i^c; \mathbf{x}_a^c)}$ and $\bar{w}_{(\mathbf{x}_i^c; \mathbf{x}_a^c)}$ respectively. The direction of each individual partial derivative is unchanged. However, since weights are rescaled non-linearly, the final partial derivative of each sample is changed to a better weighted combination of multiple partial derivatives. Final partial derivatives of $L_{\text{ICE}}(\mathbf{X}; f)$ w.r.t. positives and negatives are:

$$\frac{\partial L_{\text{ICE}}(\mathbf{X}; f)}{\partial \mathbf{f}_i^c} = -\mathbf{f}_a^c \cdot \bar{w}_{(\mathbf{x}_i^c; \mathbf{x}_a^c)} \quad \text{and} \quad \frac{\partial L_{\text{ICE}}(\mathbf{X}; f)}{\partial \mathbf{f}_j^o} = \mathbf{f}_a^c \cdot \bar{w}_{(\mathbf{x}_j^o; \mathbf{x}_a^c)}. \quad (15)$$

### 3.5 A Case Study and Intuitive Explanation of ICE

To make it more clear and intuitive for understanding, we now analyse a naive case of ICE, where there are two samples per class in every mini-batch, i.e., $\forall c, Nc = 2, |\{\mathbf{x}_i^c\}_{i \neq a}| = N_c - 1 = 1$. In this case, for each anchor (query), there is only one positive among the remaining data points. As a result, the weighting schemes in Eq. (13) for positives and Eq. (14) for negatives can be simplified:

$$\bar{w}_{(\mathbf{x}_i^c; \mathbf{x}_a^c)} = \frac{1}{2N} \cdot \frac{1 - \hat{p}(\mathbf{x}_i^c | \mathbf{x}_a^c)}{\sum_{i \neq a}(1 - \hat{p}(\mathbf{x}_i^c | \mathbf{x}_a^c))} = \frac{1}{N} \cdot \frac{1}{2}, \quad (16)$$

$$\bar{w}_{(\mathbf{x}_j^o; \mathbf{x}_a^c)} = \frac{1}{2N} \cdot \frac{\sum_{i \neq a} \hat{p}(\mathbf{x}_j^o | \mathbf{x}_a^c, \mathbf{x}_i^c)}{\sum_{i \neq a}(1 - \hat{p}(\mathbf{x}_i^c | \mathbf{x}_a^c))} = \frac{1}{N} \cdot \frac{1}{2} \cdot \frac{\hat{p}(\mathbf{x}_j^o | \mathbf{x}_a^c, \mathbf{x}_i^c)}{1 - \hat{p}(\mathbf{x}_i^c | \mathbf{x}_a^c)}. \quad (17)$$

Firstly, we have $N$ anchors that are treated equally as indicated by $1/N$. Secondly, for each anchor, we aim to recognise its positive example correctly. However, there is a *sample imbalance problem*

Table 1: A summary of three fine-grained datasets. Training and test classes are disjoint. '#' refers to the number of each item. There are only 5.3 images per class on average in SOP.

| Datasets | CARS196 | CUB-200-2011 | SOP |
|---|---|---|---|
| Context | Cars | Birds | Products |
| #Total classes | 196 | 200 | 22,634 |
| #Total images | 16,185 | 11,788 | 120,053 |
| #Training classes | 98 | 100 | 11,318 |
| #Training images | 8,054 | 5,864 | 59,551 |
| #Test classes | 98 | 100 | 11,316 |
| #Test images | 8,131 | 5,924 | 60,502 |

Table 2: The results of different reweighting parameters $s$ on SOP in terms of Recall@$K$ (%). There are 90 classes and 2 images per class in a mini-batch, i.e., the batch size is 180.

| Reweighting | R@1 | R@10 | R@100 |
|---|---|---|---|
| $s = 1$ | 42.0 | 58.1 | 74.1 |
| $s = 16$ | 71.0 | 85.6 | 93.8 |
| $s = 32$ | 73.6 | 87.5 | 94.7 |
| $s = 48$ | 76.9 | 89.7 | 95.5 |
| $s = 64$ | **77.3** | **90.0** | **95.6** |
| $s = 80$ | 75.4 | 88.7 | 94.9 |

because *each anchor has only one positive and many negatives*. ICE addresses it by treating the positive set (single point) and negative set (multiple points) equally, i.e., $1/2$ in Eq. (16) and Eq. (17) [2]. Finally, as there are many negative samples, we aim to focus more on informative ones, i.e., harder negative instances with higher matching probabilities with a given anchor. The non-linear transformation can help control the relative weight between two negative points.

The weighting scheme shares the same principle as the popular temperature-based categorical cross entropy (Hinton et al., 2015; Oreshkin et al., 2018). The key is that we should consider not only focusing on harder examples, but also the emphasis degree.

### 3.6 COMPLEXITY ANALYSIS

Algorithm 1 summarises the learning process with ICE. As presented there, the input data format of ICE is the same as CCE, i.e., images and their corresponding labels. In contrast to other methods which require rigid input formats (Schroff et al., 2015; Sohn, 2016), e.g., triplets and n-pair tuplets, ICE is much more flexible. We iteratively select one image as the anchor. For each anchor, we aim to maximise its matching probabilities with its positive samples against its negative examples. Therefore, *the computational complexity over one mini-batch is $O(N^2)$*, being the same as recent online metric learning approaches (Song et al., 2016; Wang et al., 2019b). Note that in FaceNet (Schroff et al., 2015) and $N$-pair-mc (Sohn, 2016), expensive sample mining and class mining are applied, respectively.

## 4 EXPERIMENTS

### 4.1 IMPLEMENTATION DETAILS AND EVALUATION SETTINGS

For data augmentation and preprocessing, we follow (Song et al., 2016; 2017). In detail, we first resize the input images to $256 \times 256$ and then crop it at $227 \times 227$. We use random cropping and horizontal mirroring for data augmentation during training. To fairly compare with the results reported in (Song et al., 2017), we use a centre cropping without horizontal flipping in the test phase. For the embedding size, we set it to 512 on all datasets following (Sohn, 2016; Law et al., 2017; Wang et al., 2019a). To compare fairly with (Song et al., 2017; Law et al., 2017; Movshovitz-Attias et al., 2017), we choose GoogLeNet V2 (with batch normalisation) (Ioffe & Szegedy, 2015) as the backbone architecture initialised by the publicly available pretrained model on ImageNet (Russakovsky et al., 2015). We simply change the original 1000-neuron fully connected layers followed by softmax normalisation and CCE to 512-neuron fully connected layers followed by the proposed ICE. For faster convergence, we randomly initialise the new layers and optimise them with 10 times larger learning rate than the others as in (Song et al., 2016).

We implement our algorithm in the Caffe framework (Jia et al., 2014). The source code will be available soon.

**Datasets.** Following the evaluation protocol in (Song et al., 2016; 2017), we test our proposed method on three popular fine-grained datasets including CARS196 (Krause et al., 2013), CUB-200-2011 (Wah et al., 2011) and SOP (Song et al., 2016). A summary of the datasets is given in

---

[2]The weight sum of negatives: $\sum_{o \neq c} \sum_j \hat{p}(\mathbf{x}_j^o | \mathbf{x}_a^c, \mathbf{x}_i^c) = 1 - \hat{p}(\mathbf{x}_i^c | \mathbf{x}_a^c) \implies \sum_{o \neq c} \sum_j \bar{w}_{(\mathbf{x}_j^o; \mathbf{x}_a^c)} = \bar{w}_{(\mathbf{x}_i^c; \mathbf{x}_a^c)} = 1/(2N)$.

Table 3: Comparison with the state-of-the-art methods on CARS196, CUB-200-2011 and SOP in terms of Recall@$K$ (%). All the compared methods use GoogLeNet V2 as the backbone architecture. '–' means the results which are not reported in the original paper. The best results in the first block using single embedding are bolded.

| | CARS196 | | | | CUB-200-2011 | | | | SOP | | |
|---|---|---|---|---|---|---|---|---|---|---|---|
| $K$ | 1 | 2 | 4 | 8 | 1 | 2 | 4 | 8 | 1 | 10 | 100 |
| Without fine-tuning | 35.6 | 47.3 | 59.4 | 72.2 | 40.1 | 53.2 | 66.0 | 76.6 | 43.7 | 60.8 | 76.5 |
| Fine-tuned with CCE | 48.8 | 58.5 | 71.0 | 78.4 | 46.0 | 58.0 | 69.3 | 78.3 | 51.7 | 69.8 | 85.3 |
| Triplet Semihard | 51.5 | 63.8 | 73.5 | 82.4 | 42.6 | 55.0 | 66.4 | 77.2 | 66.7 | 82.4 | 91.9 |
| Lifted Struct | 53.0 | 65.7 | 76.0 | 84.3 | 43.6 | 56.6 | 68.6 | 79.6 | 62.5 | 80.8 | 91.9 |
| N-pair-mc | 53.9 | 66.8 | 77.8 | 86.4 | 45.4 | 58.4 | 69.5 | 79.5 | 66.4 | 83.2 | 93.0 |
| Struct Clust | 58.1 | 70.6 | 80.3 | 87.8 | 48.2 | 61.4 | 71.8 | 81.9 | 67.0 | 83.7 | 93.2 |
| Spectral Clust | 73.1 | 82.2 | 89.0 | 93.0 | 53.2 | 66.1 | 76.7 | 85.3 | 67.6 | 83.7 | 93.3 |
| Proxy NCA | 73.2 | 82.4 | 86.4 | 88.7 | 49.2 | 61.9 | 67.9 | 72.4 | 73.7 | – | – |
| RLL | 74.0 | 83.6 | 90.1 | 94.1 | 57.4 | **69.7** | 79.2 | **86.9** | 76.1 | 89.1 | 95.4 |
| ICE | **77.0** | **85.3** | **91.3** | **94.8** | **58.3** | 69.5 | **79.4** | 86.7 | **77.3** | **90.0** | **95.6** |
| RLL-(L,M,H) | 82.1 | 89.3 | 93.7 | 96.7 | 61.3 | 72.7 | 82.7 | 89.4 | 79.8 | 91.3 | 96.3 |
| ICE-(L, M, H) | 82.8 | 89.5 | 93.7 | 96.4 | 61.4 | 73.2 | 82.5 | 89.2 | 80.1 | 91.8 | 96.6 |

Table 1. We also keep the same train/test splits. We remark that to test the generalisation and transfer capability of the learned deep metric, the training and test classes are disjoint.

**Evaluation protocol.** We evaluate the learned representations on the image retrieval task in terms of Recall@$K$ performance (Song et al., 2016). Given a query, its $K$ nearest neighbours are retrieved from the database. Its retrieval score is one if there is an image of the same class in the $K$ nearest neighbours and zero otherwise. Recall@$K$ is the average score of all queries.

**Training settings.** All the experiments are run on a single PC equipped with Tesla V100 GPU with 32GB RAM. For optimisation, we use the stochastic gradient descent (SGD) with a weight decay of $1e^{-5}$ and a momentum of 0.8. The base learning rate is set as $1e^{-3}$. The training converges at $20k$ iterations on SOP while $4k$ iterations on CARS196 and CUB-200-2011. As for the hyperparameters, we study their impacts in Sec. 4.3 and supplementary material. The mini-batch size is 60 for small datasets CARS196 and CUB-200-2011 while 180 for the large benchmark SOP. Additionally, we set $C = 6, N_c = 10$ on CARS196 and CUB-200-2011 while $C = 90, N_c = 2$ on SOP. The design reasons are: 1) SOP has only 5.3 images per class on average. Therefore $N_c$ cannot be very large; 2) It helps to simulate the global structure of deep embeddings, where the database is large and only a few matching instances exist.

The analysis of batch content, batch size and embedding size is presented in the supplementary material.

## 4.2 QUANTITATIVE RESULTS

**Remarks.** For a fair comparison, we remark that the methods group (Ustinova & Lempitsky, 2016; Harwood et al., 2017; Wang et al., 2017b; Duan et al., 2018; Lin et al., 2018; Suh et al., 2019; Zheng et al., 2019) using GoogLeNet V1 (Szegedy et al., 2015) and another group (Wu et al., 2017; Cakir et al., 2019; Sanakoyeu et al., 2019) using ResNet-50 (He et al., 2016) are not benchmarked. Besides, ensemble models (Yuan et al., 2017; Opitz et al., 2017; Kim et al., 2018; Xuan et al., 2018) are not considered. HTL (Ge et al., 2018) also uses GoogLeNet V2, but it constructs a hierarchical similarity tree over the whole training set and updates the tree every epoch, thus being highly unscalable and expensive in terms of both computation and memory. That is why HTL achieves better performance on small datasets but performs worse than ours on the large dataset SOP. Finally, there are some other orthogonal deep metric learning research topics that are worth studying together in the future, e.g., a robust distance metric (Yuan et al., 2019) and metric learning with continuous labels (Kim et al., 2019). In GoogLeNet V2, there are three fully connected layers of different depth. We refer them based on their depth: L for the low-level layer (inception-3c/output), M for the mid-level layer (inception-4e/output) and H for the high-level layer (inception5b/output). By default, we use only 'H'. We also report the results of their combination (L, M, H) for reference following RLL (Wang et al., 2019a).

**Competitors.** All the compared baselines, Triplet Semihard (Schroff et al., 2015), Lifted Struct (Song et al., 2016), $N$-pair-mc (Sohn, 2016), Struct Clust (Song et al., 2017), Spectral Clust (Law et al., 2017), Proxy-NCA (Movshovitz-Attias et al., 2017), RLL (Wang et al., 2019a) and our ICE are trained and evaluated using the same settings: (1) GoogLeNet V2 serves as the backbone network; (2) All models are initialised with the same pretrained model on ImageNet; (3) All apply the same data augmentation during training and use a centre-cropped image during testing. The results of some baselines (Schroff et al., 2015; Song et al., 2016; Sohn, 2016) are from (Song et al., 2017), which means they are reimplemented there for a fair comparison. In addition, the results of vanilla GoogLeNet V2 pretrained on ImageNet without fine-tuning and with fine-tuning via minimising CCE are reported in (Law et al., 2017), which can be regarded as the most basic baselines. Among these baselines, Proxy NCA is not scalable as class-level proxies are learned during training. Struct Clust and Spectral Clust are clustering-motivated methods which explicitly aim to optimise the clustering quality. We highlight that clustering performance Normalised Mutual Information (NMI) (Schütze et al., 2008) is not a good assessment for SOP (Law et al., 2017) because SOP has a large number of classes but only 5.3 images per class on average. Therefore, we only report and compare Recall@$K$ performance.

**Results.** Table 3 compares the results of our ICE and those of the state-of-the-art DML losses. ICE achieves the best Recall@1 performance on all benchmarks. We observe that only RLL achieves comparable performance in a few terms. However, RLL is more complex since it has three hyperparameters in total: one weight scaling parameter and two distance margins for positives and negatives, respectively. In addition, its perspective is different since it processes the positive set together similarly with (Triantafillou et al., 2017; Wang et al., 2019c). We note that (Wang et al., 2019c) is also complex in designing weighting schemes and contains four control hyper-parameters. However, our Recall@1 on SOP is 77.3%, which is only 0.9% lower than 78.2% of (Wang et al., 2019c). It is also worth mentioning that among these approaches, except fine-tuned models with CCE, only our method has a clear probability interpretation and aims to maximise the joint instance-level matching probability. As observed, apart from being unscalable, CCE's performance is much worse than the state-of-the-art methods. Therefore, ICE can be regarded as a successful exploration of softmax regression for learning deep representations in DML. The t-SNE visualisation (Van Der Maaten, 2014) of learned embeddings are available in the supplementary material.

### 4.3 ANALYSIS OF SAMPLE REWEIGHTING IN ICE

We empirically study the impact of the weight scaling parameter $s$, which is the only hyperparameter of ICE. It functions similarly with the popular sample mining or example weighting (Wang et al., 2019a;b;c) widely applied in the baselines in Table 3. Generally, different $s$ corresponds to different emphasis degree on difficult examples. When $s$ is larger, more difficult instances are assigned with relatively higher weights.

In general, small datasets are more sensitive to minor changes of hyper-settings and much easier to overfit. Therefore, the experiments are conducted on the large dataset SOP. The results are shown in Table 2. Note that when $s$ is too small, e.g., $s = 1$, we observe that the training does not converge, which demonstrates the necessity of weighting/mining samples. The most significant observation is that focusing on difficult samples is better but the emphasis degree should be properly controlled. When $s$ increases from 16 to 64, the performance grows gradually. However, when $s = 80$, we observe the performance drops a lot. That may be because extremely hard samples, e.g., outliers, are emphasised when $s$ is too large.

## 5 CONCLUSION

In this paper, we propose a novel instance-level softmax regression framework, named instance cross entropy, for deep metric learning. Firstly, the proposed ICE has clear probability interpretation and exploits structured semantic similarity information among multiple instances. Secondly, ICE is scalable to infinitely many classes, which is required by DML. Thirdly, ICE has only one weight scaling hyper-parameter, which works as mining informative examples and can be easily selected via cross-validation. Finally, distance thresholds are not applied to achieve intraclass compactness and interclass separability. This indicates that ICE makes no assumptions about intraclass variances and the boundaries between different classes. Therefore ICE owns general applicability.

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

# Supplementary Material for
# Instance Cross Entropy for Deep Metric Learning

## A  MORE ABLATION STUDIES

### A.1  BATCH CONTENT

We evaluate the impact of batch content which consists of $C$ classes and $k$ images per class, i.e., $\forall c, N_c = k$. The batch size $N = C \times k$ is set to 180. In our experiments, we change the number of classes $C$ from 36 to 90, and the number of images $k$ from 2 to 5, while keeping the batch size unchanged. Table 4 shows the results on SOP dataset. We observe that when there are more classes in the mini-batch, the performance is better. We conjecture that as the number of classes increases, the mini-batch training becomes more difficult and helps the model to generalise better.

Table 4: The impact of batch content $C \times k$ on SOP in terms of Recall@$K$ (%). The batch size is $N = 180$ and the scaling parameter is $s = 64$.

| $N = 180, s = 64$ | R@1 | R@10 | R@100 |
|---|---|---|---|
| $C \times k = 90 \times 2$ | 77.3 | 90.0 | 95.6 |
| $C \times k = 60 \times 3$ | 75.2 | 88.7 | 95.2 |
| $C \times k = 45 \times 4$ | 74.9 | 88.7 | 95.3 |
| $C \times k = 36 \times 5$ | 74.6 | 88.7 | 95.4 |

### A.2  BATCH SIZE

To explore different batch size $N$, we fix $k = 2$ and only change $C$. In this case, $N = C \times 2$. Table 5 shows that as the number of classes increases, the performance grows. In detail, when the number of classes increases from 50 to 90, the performance raises from 74.4% to 77.3% accordingly. One reason may be that as the number of classes increases, it fits the global structure of the test set better, where there are a large number of classes but only a few positive examples. In addition, the increasing difficulty of mini-batch training can help the model to generalise better.

Table 5: The results of different batch size $N$ on SOP in terms of Recall@$K$ (%). While changing $C$, we fix $k = 2$ and $s = 64$. Therefore, $N = C \times 2$.

| $k = 2, s = 64$ | R@1 | R@10 | R@100 |
|---|---|---|---|
| $N = 180$ | 77.3 | 90.0 | 95.6 |
| $N = 160$ | 75.4 | 88.8 | 95.1 |
| $N = 140$ | 75.1 | 88.7 | 95.2 |
| $N = 120$ | 75.1 | 88.6 | 95.2 |
| $N = 100$ | 74.4 | 88.2 | 95.1 |

### A.3  EMBEDDING SIZE

The dimension of feature representations is an important factor in many DML methods. We conduct experiments on SOP to see the influence of different embedding size. The results are presented in Table 6. We observe that when the embedding size is very small, e.g., 64, the performance is much worse. The performance increases gradually as the embedding size grows.

## B  T-SNE VISUALISATION

The t-SNE visualisation (Van Der Maaten, 2014) of learned embeddings is available in Figures 2, 3, 4.

Table 6: The results of different embedding size on SOP in terms of Recall@$K$ (%). In all experiments: $s = 64, C = 90, k = 2$. $N = C \times k = 90 \times 2$.

| $180 = 90 \times 2, s = 64$ | R@1 | R@10 | R@100 |
|---|---|---|---|
| 64 | 72.6 | 87.1 | 94.0 |
| 128 | 74.3 | 87.9 | 94.5 |
| 256 | 75.2 | 88.6 | 94.8 |
| 512 | 77.3 | 90.0 | 95.6 |

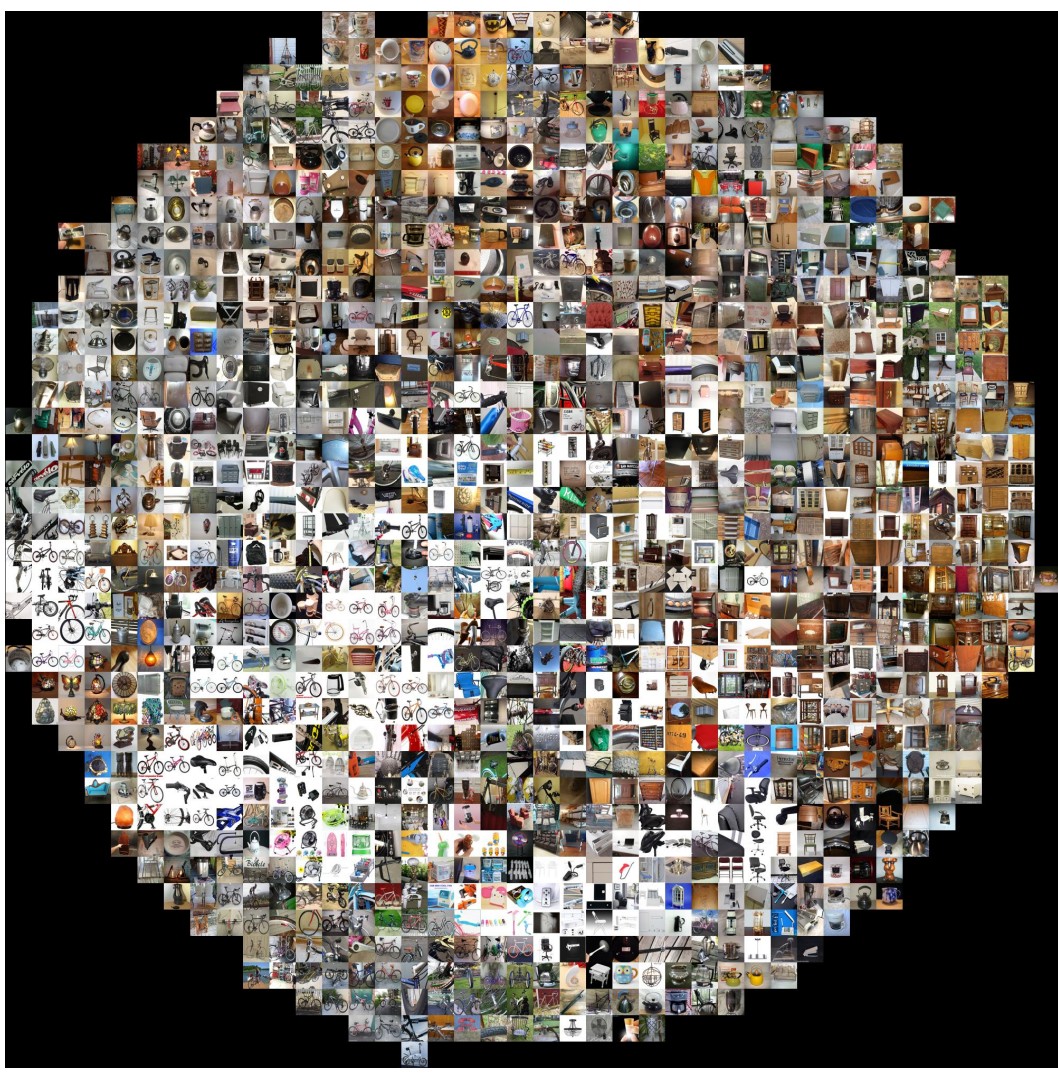

Figure 2: t-SNE visualisation (Van Der Maaten, 2014) on the SOP test set. Best viewed on a monitor when zoomed in.

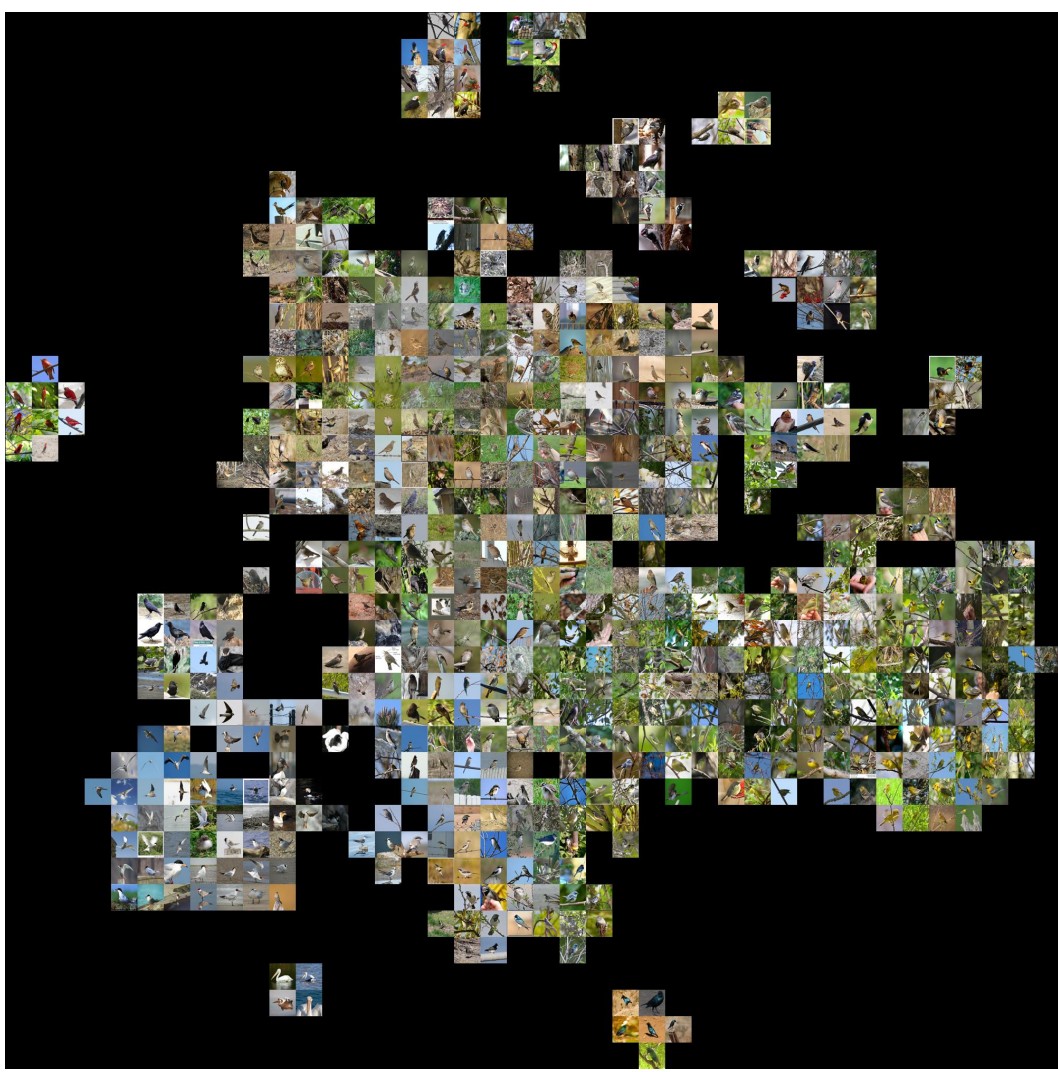

Figure 3: t-SNE visualisation (Van Der Maaten, 2014) on the CUB-200-2011 test set. Best viewed on a monitor when zoomed in.

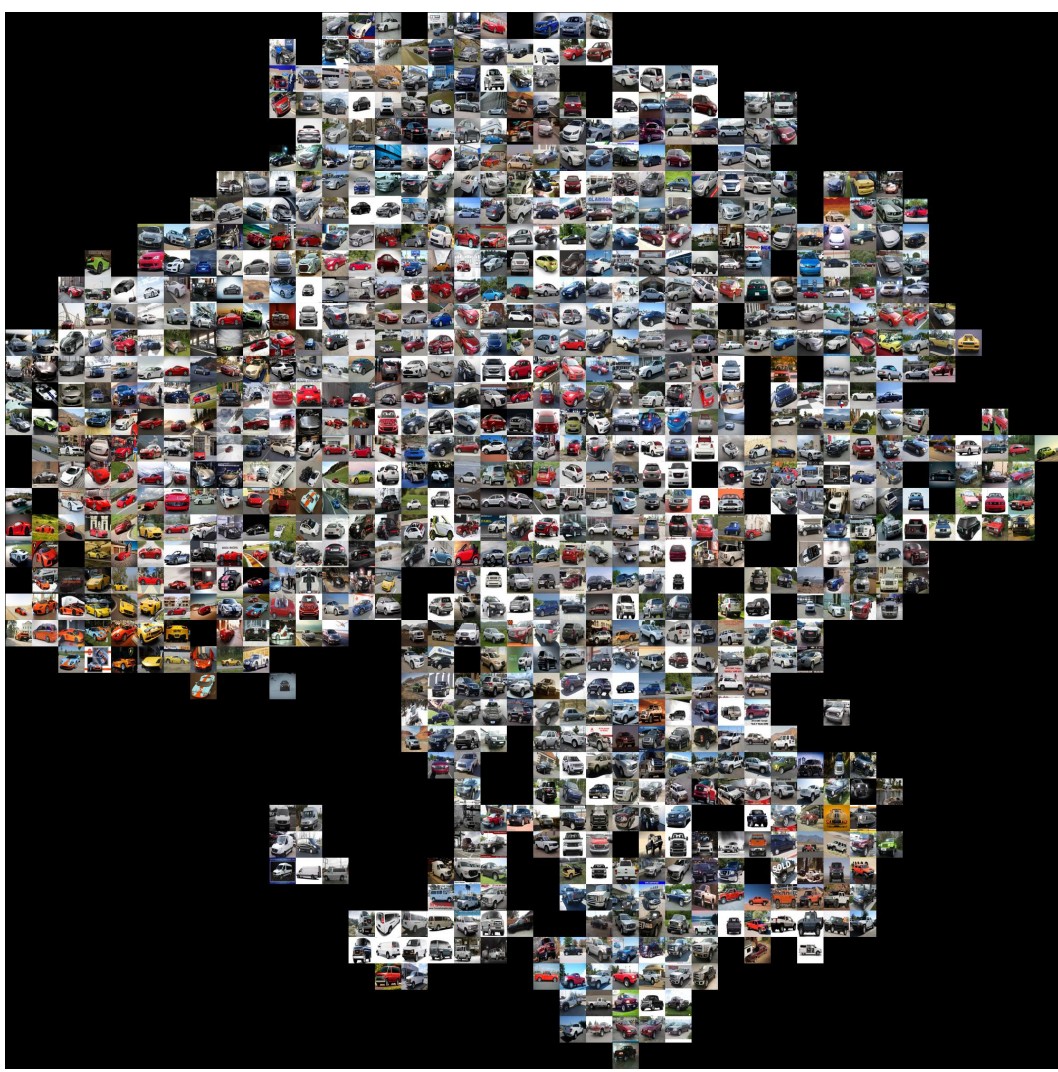

Figure 4: t-SNE visualisation (Van Der Maaten, 2014) on the CARS196 test set. Best viewed on a monitor when zoomed in.

