# OpenReview forum: "INSTANCE CROSS ENTROPY FOR DEEP METRIC LEARNING"
_ICLR.cc/2020/Conference — Reject_

### Official Review · AnonReviewer2 · 2019-10-22
**Official Blind Review #2**

**Rating:** 1

**Review:**

The paper proposes a method inspired by the categorical cross entropy (CCE). In a similar way as many metric learning approaches, a softmax approach based on cross entropy is used to enforce similar examples to have smaller distances than dissimilar distances.
Nonetheless, instead of considering a centroid-based deep metric learning approach (e.g. like Snell et al. (2017)), only one anchor is considered for each training example, and the distance of an anchor with one of its positive is learned that its distance is smaller than the distance with examples belonging to different categories.

I vote for reject for the following reasons:
- The paper is too similar to NCA and S-NCA (introduced in Section 2.2). In the same way as ICE, S-NCA also considers learning l2-regularized representations. The main difference with S-NCA is the way negative examples are sampled.
S-NCA proposes a framework based on augmented memory, ICE proposes a sampling strategy similar to [A] and used in Nickel et al. (2018) to subsample negative pairs.
- Another contribution of ICE is the use of some hyperparameter s in Equations (11) and (12). This hyperparameter s plays the role as the inverse of the temperature and is in fact learned in ICE. S-NCA also plays with the temperature but does not learn it.
On the other hand, learning the temperature in a similar way has been proposed in the deep metric learning literature (e.g. TADAM).

On the other hand, the paper has some nice contributions:
- The main "novelty" of ICE seems to be the analysis in Section 3.4 of the partial derivatives and their impact on the sample reweighting. Although the explanation is simple, it helps understand what's happening during optimization.
- The experimental results on different transfer learning benchmarks seem convincing.


[A] Jean et al. On usingvery large target vocabulary for neural machine translation. ACL 2015

**Experience Assessment:**

I have published one or two papers in this area.

**Review Assessment: Checking Correctness Of Derivations And Theory:**

I carefully checked the derivations and theory.

**Review Assessment: Checking Correctness Of Experiments:**

I assessed the sensibility of the experiments.

**Review Assessment: Thoroughness In Paper Reading:**

I read the paper at least twice and used my best judgement in assessing the paper.

---

> ### Author Response · Authors · 2019-11-06
> **The novelty and differences with prior work, NCA, S-NCA, and TADAM**
>
>
> Thanks so much for your helpful review. We are glad that you like some of our contributions.
> We read your rejection reasons carefully and they are mainly about the novelty and differences with prior work, e.g., NCA, S-NCA, and TADAM. To avoid distraction, here we would like to clarify the main differences according to Figure 1 only, which is improved in the revised version.
>
> Comparing NCA and S-NCA with ICE,  the main difference is not the way negative examples are sampled. Instead, there are two major differences when comparing Figure 1(d) and Figure 1(e):
> 1). Instance-level matching distribution: Given one query, NCA and S-NCA consider it versus the remaining positives and negatives. Consequently, there is only one instance-level matching distribution for each query.
> Contrastively, given one query, ICE considers its positives separately. As a result, the number of a query’s instance-level matching distributions equals the number of its positives.
> Therefore, you can see one instance-level matching distribution in Figure 1(d) while two instance-level matching distributions in Figure 1(e).
> 2). The definition and meaning of cross entropy are quite different: NCA and S-NCA compute the cross entropy between a predicted neighbourhood distribution and the ground-truth.
> On the contrary, in ICE, the matching distribution of a query versus each of its positives is optimised separately.
>
> In addition, original NCA learns the whole training data and its time complexity is quadratically proportional to the scale of training data. S-NCA is proposed recently with linear time complexity with respect to the training data size. Please see more discussion in the Section 2.2.
>
>
> Regarding TADAM:
> 1). Given a query, TADAM compares its similarity with class centroids as illustrated in Figure 1(b). Instead, ICE computes its similarity with other instances.
> 2). We also presented a discussion about TADAM in Section 2.4., “Remarkably, TADAM formulates instances versus class centres and also has a metric scaling parameter for adjusting the impact of different class centres. Contrastively, ICE adjusts the influence of other instances. Furthermore, ours is not  distance metric scaling since we simply apply naive cosine similarity as the distance metric at the testing stage. That is why we interpret it as a weighting scheme during training.”

---

> > ### Comment · AnonReviewer2 · 2019-11-14
> > **I still do not see the novelty**
> >
> > I have carefully read the new pdf version (November 13), and I still do not see the novelty compared existing approaches.
> >
> > 1) Comparison to (Nickel & Kiela, 2018)
> > By considering a similarity information based on category membership, the formulation of ICE in Section 3.2 can be reformulated as a softmax where $f_i^\top$ is similar to $f_a^\top$ and all the sampled $f_j^\top$ are dissimilar.
> > This formulation is very similar to the formulation of Section 3.3 of (Nickel & Kiela, 2018) where the similarity is instead based on hierarchy: concepts in the same hierarchy subtree are more similar to each other than to concepts of other subtrees.
> > By using a sampling strategy inspired by [A], dissimilar examples are subsampled to provide efficient optimization.
> > The optimization scheme in (Nickel & Kiela, 2018) is similar to ICE except that the similarity information is based on hierarchy instead of categories.
> >
> > A study of gradients, similar to the one in ICE, is provided in Section 3 of [A].
> >
> > 2) Comparison to (Snell et al., 2017)
> > ProtoNets also have the same kind of softmax optimization, where instead of considering a single similar example, a similar prototype (computed as the average of similar examples) is used.
> > The number of dissimilar constraints (used to normalize the softmax) is also smaller since one prototype is used to represent the distance of a sample to dissimilar categories.
> > (Wang et al., 2017a / FaceNet) have an optimization scheme similar to (Snell et al., 2017) except that cosine similarity is used, and a study of gradients is provided in the appendix of (Wang et al., 2017a / FaceNet).
> >
> >
> >
> > [A] Jean et al. On using very large target vocabulary for neural machine translation. ACL 2015

---

> > > ### Author Response · Authors · 2019-11-14
> > > **Differentiate our work from prior work**
> > >
> > >
> > > Thank you for your reply and new reviews.
> > > We find some misunderstandings exist. Therefore, we address your concerns as follows:
> > >
> > > 1. Comparison to (Nickel & Kiela, 2018)
> > > Hyperbolic (Nickel & Kiela, 2018) is different from ours. We have a discussion on this in Section 2.2. The details are “ It aims to preserve the similarity structures among instances as well. However, it learns a hyperbolic embedding space where the distance depends only on norm of embeddings. Instead, we learn an angular space where the similarity depends only on the angle between embeddings. Besides, Hyperbolic requires a separate sampling of semantic subtrees when the dataset is large.”
> > >
> > >
> > > 2. [A] is unrelated to our work. We guess you made a mistake here. “[A] Jean et al. On using very large target vocabulary for neural machine translation. ACL 2015”.
> > > You wrote: (1) “By using $\mathrm{a \ sampling \ strategy}$ inspired by [A], dissimilar examples are subsampled to provide efficient optimization”; (2) “A study of gradients, similar to the one in ICE, is provided in Section 3 of [A].”
> > > We checked this paper carefully, especially its methodology Sections 2 and 3. To address your concerns, we present $\mathrm{Our \ Findings}$:
> > >   - We have no sampling process. [A] proposed $\mathrm{a \ sampling \ strategy}$ to reduce the computational complexity when predicting the probability of the next target word. As a result, a subset of target words are sampled by importance sampling. Another consequence is that the output probability becomes approximated. $\mathrm{That \ is \  because \  when  \ conducting \  softmax  \ normalisation,  \ only  \ a  \ subset  \ of  \ the  \ target  \ vocabulary  \ is \  used  \ in \  the \  denominator. }$    However, we provide a discussion in our Section 2.3 that our method does not apply sampling strategies.
> > >   - [A] approximately predicts the probability of the next target word during training. Only a subset of target vocabulary is subsampled, which means the target space (vocabulary space) is smaller after sampling.
> > >   - The gradient study is different: the gradient analysis in [A] is for the approximation purpose with a small subset of target vocabulary. While our analysis is for introducing why we need non-linear scaling to control the relative weight between two examples.
> > >
> > >
> > > 3.  Comparison to (Snell et al., 2017)
> > > We represent Prototypical Networks (Snell et al., 2017) in Section 2.1 and Figure 1. In Prototypical Networks, a class centre is the mean over all embeddings of a class. By comparing a sample with other examples other than class centres, more informative instances can contribute more in ICE.
> > >
> > >
> > > 4. Comparison to NormFace (Wang et al., 2017a), FaceNet (Schroff et al., 2015)
> > > NormFace (Wang et al., 2017a) is discussed in Section 2.1 and Figure 1.
> > > FaceNet (Schroff et al., 2015) is introduced Section 1 and 2.3.
> > >
> > > We hope our replies address your concerns. Thank you again for your reviews!

---

### Official Review · AnonReviewer3 · 2019-10-23
**Official Blind Review #3**

**Rating:** 8

**Review:**

The paper proposes a method to measure the difference between an
estimated instance-level matching distribution and its ground-truth
one, based on instance cross entropy (ICE).
The goal is to learn an embedding that captures the semantic
similarities among samples.
In particular, with ICE they try to maximize the matching probability
of an instance with similar instances (same class).
The authors also use sample re-weighting into ICE and show the benefits
of the approach against other state-of-the-art methods on three
datasets. The authors performed several experiments with convincing
results.

The positive aspect of the paper is introducing this instance-based
measure which is shown to perform well on 3 challenging datasets.

It is not clear how is the scaling parameter (s) determined. Are there
any guidelines for fixing its value?
The authors should explain in more detail how is the non-linear
transformation achieved.

The algorithm goes through all the examples of a class on each
mini-batch, which seems a computational expensive procedure. The paper
will benefit if time results are reported for different approaches.

The paper is sometimes difficult to follow and needs a careful
revision.


**Experience Assessment:**

I do not know much about this area.

**Review Assessment: Checking Correctness Of Derivations And Theory:**

I assessed the sensibility of the derivations and theory.

**Review Assessment: Checking Correctness Of Experiments:**

I assessed the sensibility of the experiments.

**Review Assessment: Thoroughness In Paper Reading:**

I read the paper at least twice and used my best judgement in assessing the paper.

---

> ### Author Response · Authors · 2019-11-08
> **Setup of the scaling parameter and complexity analysis**
>
>
> 1. It is not clear how is the scaling parameter (s) determined.  Are there any guidelines for fixing its value?
> The authors should explain in more detail how is the non-linear transformation achieved.
>
> In our experiments, we choose and fix the scaling parameter $s$ empirically. We have presented an ablation study of $s$ in Table 2. We will study your questions in our future work: 1). How to choose $s$ automatically, e.g., using AutoML techniques to optimise this hyper-parameter $s$; 2). How to change $s$ dynamically as training goes instead of fixing it.
>
> As represented in Eq. (11) and (12), we scale the absolute weight non-linearly, i.e., linear scaling followed by the non-linear exponential function. Non-linear transformation of the absolute weight leads to the change of relative weight, which is an indirect way of controlling the relative weight.
>
>
> 2. The algorithm goes through all the examples of a class on each mini-batch, which seems a computational expensive procedure. The paper will benefit if time results are reported for different approaches.
>
> We presented a complexity analysis in Section 3.6. We highlighted that: 1). ICE does not require rigid input formats, thus being faster and more flexible in terms of mini-batch data preparation; 2). In addition, ICE does not need expensive sample mining or class mining; 3). The computational complexity over one mini-batch is $O(N^2)$, being the same as recent metric learning approaches. In our implementation, we only compute once the similarities between every two instances and store them to reuse multiple times when computing ICE.
> We do not report the time results of different approaches because some of them are implemented in different frameworks, making it not easy to reimplement and conduct a fair comparison.
>
> 3. Our representation is improved in the revised version.

---

### Official Review · AnonReviewer4 · 2019-11-12
**Official Blind Review #4**

**Rating:** 3

**Review:**

Overview

This paper proposes a new objective function called Instance Cross Entropy (ICE) for metric learning. Compared to the triplet loss (or contrastive loss) and its many variants, the distance between points in feature space is defined to be the dot/inner product, rather than computing the euclidian distance between feature vectors. Since the L2 norm of the final features is constrained  to be 1, the dot product represents the cosine distance between 2 feature vectors. Compared to the softmax/categorical loss, this objective has the advantage that there is no need to learn a per-class weight vector in the output softmax layer and therefore this method can be used even when the number of classes during training is unbounded. This is a very useful design feature, since the output softmax layer grows linearly with the size of the classes and can quickly become prohibitively large. Furthermore, mini batches for training can be randomly sampled without requiring expensive negative data mining strategies which are necessary when using the triplet loss. This is another very useful property.

Overall, I think the ideas presented are interesting and the paper provides a useful summary of existing approaches for metric learning. However, I think the technical presentation needs to be improved significantly before  the paper can be accepted for publication. I found the paper quite difficult to follow. I had to re-read the motivation and the description of the loss function many times before being able to understand what is going on and how the proposed loss compares and contrasts with existing approaches. I also think some of the terminology and notation used in the paper is very confusing and should be updated to help the reader get to the main argument quickly.

Comments


1. Title: What exactly does instance cross entropy (ICE) mean? And how is it different from categorical cross entropy (CCE)? Isn’t categorical cross-entropy also calculating the entropy between the predicted distribution for *each instance* and the ground truth? I think the authors should reconsider the name for the proposed loss and choose a more descriptive name for the algorithm.

2.  I also found the use of the term “matching distribution” a bit confusing since it has a well-defined meaning in statistics. However ML is full of overloaded terms and I understand if the authors want to keep this description.

3. None of the references in the main text have brackets around them. I think this makes the paper appear very cluttered and makes parsing each paragraph quite difficult. I would highly recommend the paper be reformatted and the authors use brackets around references.

4. In the abstract, the authors mention that the proposed method has a “clear probabilistic interpretation”. From my understanding, I see ICE as a blend of the categorical cross-entropy loss and the triplet loss, retaining useful properties of each without increasing the complexity of the loss computation. However, I do not see a clear probabilistic interpretation of the loss function. The softmax computation yields a probability distribution like output given a query and an anchor point, which is hand-designed in the objective function. I  think the authors should more clearly motivate what the probabilistic angle for the loss function is.

5. I found Figure 1 to be very difficult to interpret. This is a missed opportunity, since this figure alone could communicate some of the key ideas in the paper to the reader. However, there are many details that have not been explicitly mentioned. What do the colours (white, blue, yellow) mean? The figure shapes probably mean classes. What do the terms p_* mean? What do i,j mean in Figure 1 e and how are they different from the previous figures. These details should all be present in the title for the figure. Even with these details, the figure could still be a bit more explicit. I found the yellow arrow with cross entropy and the notion of “ground truth” in the figure difficult to follow.

6. How was the constrained optimisation performed? In Equation 6, we see that there is a constraint associated with each example in the mini batch which says the norm of the feature vector should be equal to 1. However I could not find any implementation details of how this constraint was satisfied. Did the authors use a Lagrangian formulation? I think the paper is irreproducible without this detail.

Summary

I think this is an interesting proposal to combine the useful features of the softmax and triplet losses. However, I think the technical presentation needs to be improved significantly in order for the paper to be accepted for publication.


**Experience Assessment:**

I have published one or two papers in this area.

**Review Assessment: Checking Correctness Of Derivations And Theory:**

I carefully checked the derivations and theory.

**Review Assessment: Checking Correctness Of Experiments:**

I assessed the sensibility of the experiments.

**Review Assessment: Thoroughness In Paper Reading:**

I read the paper thoroughly.

---

> ### Author Response · Authors · 2019-11-13
> **The ideas presented are interesting and the paper provides a useful summary of existing approaches for metric learning. However, I think the technical presentation needs to be improved significantly before  the paper can be accepted for publication.**
>
>
> 1. Title: What exactly does instance cross entropy (ICE) mean? And how is it different from categorical cross entropy (CCE)?
>
> Thanks, we introduce why we choose the name, instance cross entropy (ICE), as follows:
> - Categorical cross entropy (CCE) is the cross entropy between: (1) a predicted matching distribution of $\mathit{one \ query \ versus \ class\text{-}level \ weight \ vectors, \ including \ one \ positive \ class \ centroid \ and \ multiple \ negative \ ones}$, and (2) its corresponding ground-truth.
> - Instance cross entropy is the cross entropy between: (1) a predicted matching distribution of $\mathit{one \ query \ versus \ other \ instances, \ including \ one \ positive \ and \ multiple \ negatives}$, and (2) its corresponding ground-truth.
>
> Therefore, following the well-known name of $categorical$ cross entropy where ‘categorical’ indicates matching a query with class-level centroids, we name our method instance cross entropy where ‘instance’ indicates matching a query with instances.
> The name is chosen in a symmetric way to categorical cross entropy.
>
>
> 3. None of the references in the main text have brackets around them.
>
> Thanks so much for your recommendation. We have put the citations into parentheses in the revised version.
>
>
> 4. In the abstract, the authors mention that the proposed method has a “clear probabilistic interpretation”. …… The authors should more clearly motivate what the probabilistic angle for the loss function is.
>
> - From the probabilistic angle, ICE aims to minimise the cross entropy between: (1) a predicted matching distribution of one query versus other instances (one positive instance and multiple negative ones), and (2) its corresponding ground-truth (One-hot representation).
>
> - Regarding the probabilistic interpretation, a predicted matching distribution is composed of probabilities of a query matching a positive instance and this query matching different negative instances. Our intuitive motivation is to maximise the probability of a query matching a positive instance while simultaneously minimise the probability of this query matching any of its negative instances.
>
> - In terms of mathematical modelling and formulation, ICE is consistent with CCE. However, the interpretation differs. CCE maximises the probability of an example matching its corresponding ground-truth class, while at the same time minimises the probability of it matching any other class. In CCE, a class is represented by a learned parametric weight vector, whose dimension is the same as the learned representations of instances.
>
>
> 5. I found Figure 1 to be very difficult to interpret. This is a missed opportunity, since this figure alone could communicate some of the key ideas in the paper to the reader.
>
> In the revised version, our figure is improved significantly according to your suggestions. Thank you so much.
>
>
> 6. How was the constrained optimisation performed?
>
> The $L_2$-normalisation layer is implemented according to NormFace (Wang et al., 2017a). We add this information to our implementation details of the revised version. We will also release our code soon.

---

> > ### Comment · AnonReviewer4 · 2019-11-14
> > **Response to Authors**
> >
> > Thanks to the authors for making the suggested changes so quickly!
> >
> > 1. Thanks for putting brackets around the references, the text is significantly easier to read.
> >
> > 2. Thank you updating the figure! It is so much clearer now and so much more informative. I was able to get the gist of the paper from this figure alone.
> >
> > 3. Regarding the $L_2$-normalisation layer and the NormFace (Wang et al., 2017a) paper
> >
> > a. Section 3.3 claims that feature normalisation should be regarded as a form of regularisation to prevent the norms of inner products becoming very large, however most cross-entropy methods do not apply normalisation to the weights or input features and gradient descent seems to work fine in this setting. Do the authors have a comment on this? For example any neural net with an output softmax layer is trained without feature/weight normalisation without the parameter values diverging.
> >
> > b. The authors should cite the (Wang et al., 2017a) paper in Section 3.3 and explicitly mention that the L2-normalisation layer has the same formulation  as this paper. Maybe even mention that this is  a differentiable operation and can be easily inserted at the output of the neural net.
> >
> > c. The authors claim that one of the main features of the proposed loss function is the probabilistic nature of the loss function. However, having the re-read the paper and the Wang et al., 2017a paper on normalisation, I have a few doubts and questions regarding this interpretation. With the help of the L2-normalisation layer, the outputs of the neural network are constrained to have unit norm and therefore inner products represent cosine distances. When the norm is constrained to be 1, $\|x-y\| = 2 - x^Ty$. Therefore from Equation 3 in the paper, isn't $ (f^c_a)^T f^c_i \propto \|f^c_a- f^c_i\| \propto d(f^c_a, f^c_i) $?  Given this formulation, isn't it more natural to view this loss as a variant of the triplet loss? Equation 3 can now be re-written as: $$C(x^c_a,x^c_i) = \frac{\exp({d(f^c_a,f^c_i))}}{\exp({d(f^c_a,f^c_i))} + \sum _{o \neq c}\sum _j\exp({d(f^c_a,f^o_j))}},$$ where rather than directly optimising the Euclidean distances between the positive, anchor and negative example, there is now an additional softmax normalisation step. The authors should comment on how this formulation is related to the triplet loss and the n-pair-mc loss and maybe also clarify the probabilistic interpretation given that we are computing distances.

---

> > > ### Author Response · Authors · 2019-11-14
> > > **3. Regarding the normalisation layer and the NormFace (Wang et al., 2017a) paper**
> > >
> > >
> > > Thank you for your reply and new reviews.
> > >
> > > 3.a. Comments on the feature normalisation.
> > > It is true that traditionally, CCE is trained naively without feature/weight normalisation.
> > > Recently, normalisation for weights parameters and output representations becomes popular and is widely studied. It has been demonstrated that those normalisation techniques can generally improve the stability of training process, the convergence speed of training, and the performance of learned models.
> > >
> > > In our practice, with $L_2$ normalisation of features, training is easy and convergence is fast. Without it, we have to set the learning rate to be very small and convergence is significantly slower. Finally, the performance is also worse.
> > >
> > > In addition, the following are some examples of prior work:
> > > - Normalisation for weights parameters, e.g., weight normalisation (Salimans et al., NeurIPS 2016) and SphereFace (Liu et al., CVPR 2017).
> > > - Normalisation for both weights and features, which is widely applied in face community. For example, NormFace (Wang et al., Multimedia 2017), CosFace (Wang et al., CVPR 2019) and ArcFace (Deng et al., CVPR 2019). NormFace provides some good empirical analysis on the difference between using normalisation and without using normalisation.
> > > - Normalisation for features only, which is popular in recent deep metric learning community. For example, S-NCA, N-pair-mc and RLL of our compared baselines.
> > >
> > > References:
> > > Weight Normalization: A Simple Reparameterization to Accelerate Training of Deep Neural Networks, Salimans et al., NeurIPS 2016.
> > > SphereFace: Deep Hypersphere Embedding for Face Recognition, Liu et al., CVPR 2017.
> > > NormFace: L2 Hypersphere Embedding for Face Verification, Wang et al., ACM Multimedia 2017.
> > > CosFace: Large Margin Cosine Loss for Deep Face Recognition, Wang et al, CVPR 2018.
> > > ArcFace: Additive Angular Margin Loss for Deep Face Recognition, Deng et al, CVPR 2019.
> > >
> > >
> > > 3.b. The authors should cite the (Wang et al., 2017a) paper in Section 3.3 and explicitly mention that the L2-normalisation layer has the same formulation  as this paper. Maybe even mention that this is  a differentiable operation and can be easily inserted at the output of the neural net.
> > >
> > > Many thanks. We improved it in the new revised version according to your suggestions.
> > >
> > >
> > > 3.c. We suppose there is some misunderstanding and we would like to clarify it.
> > > - After feature $L_2$ normalisation, $\mathbf{\mathrm{inner \ product \  represents \  cosine \  similarity \  instead \  of \  distance}}$. Therefore,  $$ (\mathbf{f}^c_a)^T \mathbf{f}^c_i \propto \frac{1}{\|\mathbf{f}^c_a- \mathbf{f}^c_i\|}  \propto \frac{1}{d(\mathbf{f}^c_a, \mathbf{f}^c_i)} \propto {sim(\mathbf{f}^c_a, \mathbf{f}^c_i)},$$
> > > where $sim$ means similarity metric.  Equation (3) can now be reformulated as:
> > >  $$p(\mathbf{x}^c_i|\mathbf{x}^c_a) = \frac{\exp(1/{d(\mathbf{f}^c_a,\mathbf{f}^c_i))}}{\exp({1/d(\mathbf{f}^c_a,\mathbf{f}^c_i))} + \sum _{o \neq c}\sum _j\exp({1/d(\mathbf{f}^c_a,\mathbf{f}^o_j))}}= \frac{\exp({sim(\mathbf{f}^c_a,\mathbf{f}^c_i))}}{\exp({sim(\mathbf{f}^c_a,\mathbf{f}^c_i))} + \sum _{o \neq c}\sum _j\exp({sim(\mathbf{f}^c_a,\mathbf{f}^o_j))}},$$
> > > where $p(\mathbf{x}^c_i|\mathbf{x}^c_a)$ represents the probability of $\mathbf{x}^c_i$ matching $\mathbf{x}^c_a$ against the negative set $\{\mathbf{x}^o_j\}_{o \neq c}$. The probabilistic interpretation is in a symmetric way to CCE in Eq. (1):
> > > $$p({\mathbf{w}_{y_i} | \mathbf{x}_i}) = \frac{\exp(\mathbf{f}_i^\top\mathbf{w}_{y_i})}{\sum\nolimits_{k=1}^T \exp(\mathbf{f}_i^\top\mathbf{w}_k)} = \frac{\exp(\mathbf{f}_i^\top\mathbf{w}_{y_i})}{\exp(\mathbf{f}_i^\top\mathbf{w}_{y_i})+\sum\nolimits_{k \neq y_i} \exp(\mathbf{f}_i^\top\mathbf{w}_k)},$$
> > > where dot product represents the similarity between a query’s representation $\mathbf{f}_i$ and class-level weight vectors $\{\mathbf{w}_k\}_{k=1}^T$.
> > >
> > > In summary, the probabilistic interpretation of Eq. (1) represents a query matching its ground-truth class’s weight vector while that of Eq. (3) represents a query matching a positive. Here, the matching probability is computed by  dot product (similarity metric of two vectors) followed by softmax normalisation.
> > >
> > >
> > > - Regarding triplet loss, the definition is:
> > > $$L_{\mathrm{Triplet}}(\mathbf{x}^c_a, \mathbf{x}^c_i, \mathbf{x}^o_j;d) = \max(0, d(\mathbf{x}^c_a, \mathbf{x}^c_i) + {margin} - d(\mathbf{x}^c_a, \mathbf{x}^o_j)).$$
> > > Given a query/anchor $\mathbf{x}^c_a$, only one positive $\mathbf{x}^c_i$ and one negative $\mathbf{x}^o_j, o\neq c$ are considered. And a predefined distance margin is required.
> > >
> > > - We discuss the triplet loss (FaceNet, Schroff et al., 2015) and N-pair-mc in Section 1 and 2, respectively. We clarify the probabilistic interpretations of Eq (1) and (3) in our new revised version.
> > >
> > >
> > > We hope our replies address your concerns. Thank you again for your reviews!

---

### Decision · Program_Chairs · 2019-12-19

**Decision:**

Reject

**Comment:**

The paper proposes a new objective function called ICE for metric learning.

There was a substantial discussion with the authors about this paper. The two reviewers most experienced in the field found the novelty compared to the vast existing literature lacking, and remained unconvinced after the discussion. Some reviewers also found the technical presentation and interpretations to need improvement, and this was partially addressed by a new revision.

Based on this discussion, I recommend a rejection at this time, but encourage the authors to incorporate the feedback and in particular place the work in context more fully, and resubmit to another venue.

---

> ### Author Response · Authors · 2020-02-16
> **Its application in SimCLR-A Simple Framework for Contrastive Learning of Visual Representations**
>
>
> Hi everyone, I am Xinshao Wang, I am very glad to highlight that:  our proposed ICE is simple and effective, which has also been demonstrated in recent work SimCLR, in the context of self-supervised learning:
> A Simple Framework for Contrastive Learning of Visual Representations
>
> Its loss expression NT-Xent (the normalized temperature-scaled cross entropy loss) is a fantastic application of our recently proposed Instance Cross Entropy for Deep Metric Learning,  in the context of self-supervised learnining
>
> I am very excited about this.
>
> #InstanceCrossEntropy #TemperatureScaling #RepresentationLearning
> https://arxiv.org/pdf/1911.09976.pdf
> https://www.reddit.com/r/MachineLearning/comments/f4x1sh/r_instance_cross_entropy_for_deep_metric_learning/